# One-Step Photochemical Immobilization of Aptamer on Graphene for Label-Free Detection of NT-proBNP

**DOI:** 10.3390/bios12121071

**Published:** 2022-11-23

**Authors:** Nikita Nekrasov, Anastasiia Kudriavtseva, Alexey V. Orlov, Ivana Gadjanski, Petr I. Nikitin, Ivan Bobrinetskiy, Nikola Ž. Knežević

**Affiliations:** 1Center for Probe Microscopy and Nanotechnology, National Research University of Electronic Technology, Moscow, 124498 Zelenograd, Russia; 2Prokhorov General Physics Institute of the Russian Academy of Sciences, 119991 Moscow, Russia; 3BioSense Institute—Research and Development Institute for Information Technologies in Biosystems, University of Novi Sad, 21000 Novi Sad, Serbia

**Keywords:** graphene biosensor, heart failure, field-effect transistor, point-of-care diagnostic, aptamer, azide modification, photochemistry

## Abstract

A novel photochemical technological route for one-step functionalization of a graphene surface with an azide-modified DNA aptamer for biomarkers is developed. The methodology is demonstrated for the functionalization of a DNA aptamer for an N-terminal B-type natriuretic peptide (NT-proBNP) heart failure biomarker on the surface of a graphene channel within a system based on a liquid-gated graphene field effect transistor (GFET). The limit of detection (LOD) of the aptamer-functionalized sensor is 0.01 pg/mL with short response time (75 s) for clinically relevant concentrations of the cardiac biomarker, which could be of relevance for point-of-care (POC) applications. The novel methodology could be applicable for the development of different graphene-based biosensors for fast, stable, real-time, and highly sensitive detection of disease markers.

## 1. Introduction

B-type natriuretic peptide (BNP) has an important role in regulating circulation by acting on blood vessels in response to wall stress. The biologically active BNP is being cleaved from proBNP upon its secretion, in a process that also produces a biologically inactive N-terminal proBNP [1]. Compared to BNP with a half-life of about 20 min, NT-proBNP poses a longer half-life of 1–2 h, providing more effective target for ventricular status diagnostics. [2,3]. Thus, NT-proBNP is considered as a promising heart failure marker for point-of-care (POC) diagnostic applications. POC devices containing functionalized antibodies for NT-proBNP detection have been developed so far with electrochemical [4] or colorimetric [5] signaling. However, the use of antibodies is expensive, while rapid, accurate, and early diagnostics of heart disease remains challenging. Recently, a new aptamer with high affinity to NT-proBNP was discovered [6] and its efficient use in development of biosensors was demonstrated [7,8]. The use of aptamers as bioreceptors is a prospective method for the development of low-cost aptasensors [9,10], which can also be improved by photochemical modification of carbon lattice [11].

Conventional methods for aptamer immobilization are based on π-π stacking of pyrene-based linkers, which require multiple steps for receptor layer binding. In addition, due to the noncovalent nature of functionalization, these biosensors may suffer from instability and fast degradation of characteristics during multiple washings and measurements [9,12]. The use of photochemistry for direct attachment of bioreceptors to carbon nanomaterials is a promising technology for large-scale and reproducible fabrication of biosensors. Recently, we demonstrated the direct method of covalent functionalization of carbon nanotube transistors with proteins using phenyl azide photochemistry [13]. The photochemistry of different azide derivatives has been studied for the last 50 years [14]. Compared to actively used diazirine photochemistry, the azide introduction to biological molecules is less time-consuming and its binding to C=C bonds requires only a single wavelength source [15]. However, it is known that the intense UV irradiation can cause damage to DNA-based bioreceptors, while complex three-step click chemistry is needed for the photochemical functionalization of graphene [16]. The photochemical attachment of biological molecules to a graphene surface is a challenging task. This methodology can find widespread application potential, such as in devising new biosensors and biorobots, including elements of theragnostics and biocomputing [17]. 

In this work, we demonstrate the principle for the construction of biosensors, through the photochemical covalent binding of aptamers to graphene channels of a field effect transistor (GFET). The novel biosensor is developed to analyze NT-proBNP as a biomarker for the early detection of heart failure. Previously, GO-FET modified with an NT-proBNP antibody was successfully used to detect a biomarker with a quantified limit of detection (LOD) of 10 pg/mL [18]. Herein, we report that an azide-modified NT-proBNP aptamer can be successfully immobilized photochemically onto the graphene transducer channel as a surface receptor, to construct a biosensor for the biomarker with a quantified LOD of 0.01 pg/mL. The functionalized GFET is thoroughly investigated to prove that the aptamer is sufficiently stable even under the applied conditions of UV irradiation, to enable the specific binding of the NT-proBNP to the functionalized GFET. The sensing capabilities of the GFET in physiological solution is investigated in a wide range of NT-proBNP concentrations, while the specificity of detection is proven in the presence of another cardiac biomarker, cardiac troponin I (cTnI).

## 2. Materials and Methods

### 2.1. Materials

Phosphate buffer tablets (PBS, 137 mM NaCl, 2.7 mM KCl, and 10 mM phosphate) and Tween were purchased from Sigma-Aldrich (Burlington, MA, USA). The 3′-azide-modified NT-proBNP aptamer (5′-NH2-GGCAGGAAGACAAACAGGTCGTAGTGGAAACTGTCCACCGTAGACCGGTTATCTAGTGGTCTGTGGTGCTGT-N3-3′) was purchased from DNA Synthesis, LLC (Moscow, Russia) and purified using denaturing polyacrylamide gel electrophoresis (PAGE) to reach a mass of 3.1 nM (69.5 µg). The azide group was introduced in the oligonucleotide during the synthesis using azide-modified controlled pore glass (CPG). Frozen solutions of 1.18 mg/mL of NT-proBNP in 10 mM potassium phosphate, pH 7.4, 150 mM NaCl, and 0.7 mg/mL of cTnI in 10 mM HCl were purchased from HyTest Ltd. (Turku, Finland).

### 2.2. Graphene Surface Photochemical Modification

GFET chips were provided by Dr. Dmitry Kireev (The University of Texas at Austin, Austin, TX, USA) and metal feedlines were covered by a passivation layer to prevent current leakage during measurements in a liquid (Figure 1a). More details on GFET preparation can be found elsewhere [9]. To perform the liquid gate measurements in a solution droplet, a 5 mm diameter well punched in PDMS was placed on the chip to prevent solvent leakage during measurement. 

Azide-modified aptamer was dissolved in 1 mL of 1 × PBS to achieve 3 μM at a room temperature [19]. A 100 µL drop was placed on GFET prior to binding. The photochemical binding of the azide-modified aptamer was performed under irradiation with a UV diode (310 nm, 7–8 mW) placed at the distance of 1 cm above the GFET for 10 min [20]. After washing with DI water, the graphene FETs were left to dry. 

Atomic force microscopy (AFM, Solver-PRO, NT-MDT, Moscow, Russia) was used to estimate the change of thickness and surface roughness of graphene channel after aptamer binding. The graphene surface before and after aptasensor assembling was investigated by microRaman spectroscopy (Centaur HR, Nanoscan Technology, Dolgoprudny, Moscow Oblast, Russia) with a 100× objective (NA = 0.9) at a 532 nm wavelength (Cobolt, Solna, Sweden) with a beam spot of 1 µm^2^ and laser power of 0.5 mW.

### 2.3. Electrical Sensing 

Electrical measurements were conducted using a homemade probe station with semiconductor parameter analyzer IPPP1/5 (MNIPI, Minsk, Belarus). For liquid gate measurements, Ag/AgCl pellet electrode (Science Products GmbH, Hofheim am Taunus, Germany) with a diameter of 1 mm was introduced in the PDMS well containing the cardio marker (CM) solution. The NT-proBNP was dissolved in 0.1 × PBS with addition of tween (0.01%) to avoid sticking of proteins to the tube wall. Solutions of CM were prepared at different concentrations (0.1 pg/mL, 1 pg/mL, 10 pg/mL, 100 pg/mL, 1000 pg/mL, 5000 pg/mL, and 10000 pg/mL) to cover the range of its clinical significance [18]. The measurements were performed from the solutions in order from the lowest to the highest concentration of CM. The drop of NT-proBNP solution was placed in a PDMS well and the measurements were performed. After the measurements, each drop was replaced with the drop of solution with higher concentration of CM.

## 3. Results and Discussion

### 3.1. Photochemical Attachment of Azide-Modified Aptamers to Graphene Surface

The azide-based covalent modification of carbon nanomaterials was demonstrated previously for both carbon nanotubes [21] and graphene [22] using the high temperature treatment up to 150 °C. However, high temperatures are not suitable for the attachment of aptamers due to degradation of biomolecules. On the other hand, the UV degradation rate of DNA is a function of irradiation wavelength, with the observed degradation even at a dose of 0.1 kJ/m^2^ at 254 nm irradiation, while staying stable at a dose of 120 kJ/m^2^ in the case of irradiation at 365 nm UV [23]. Recent results showed that the irradiation of aptamers with 266 nm UV causes lesions with a relatively low damage rate, with a decay constant of about 0.4–0.8 h^−1^, for an exposure rate at 2 mJ cm^−2^ s^−1^ [24]. Furthermore, it was demonstrated that a portion of aptamers remained functionally intact even after prolonged UV irradiation. 

Hence, it can be considered that irradiation at 310 nm for up to 10 mW/cm^2^ during 10 min irradiation might still yield an FET-surface functionalization, with a sufficient portion of functionally intact aptamers to enable the desired biosensing of NT-proBNP. 

The GFET channel consists of a single layer of graphene with randomly distributed bilayer islands (Figure 1a). Upon UV irradiation of GFET in the presence of the aptamer, we observed a left shift of the Dirac point (Figure 1b). The shift reached a value of up to −48 mV for 10 min UV irradiation. Azide itself does not cause a significant shift of the Dirac point [13] because only one electron can be transferred during nitrene formation (Figure 1c). The major impact on electrical properties can be observed after aptamer binding, as the backbone brings negative charges close to the carbon lattice [9,25]. The Dirac shift for UV treatment of graphene in the same conditions (in PBS) but without the aptamer led to a minor shift of the Dirac point (Appendix A), which can be ascribed to the partial oxidation of organic residuals on the top of the graphene surface [26]. No changes in I-V_G_ curves were observed in the presence of aptamers without exposure to UV, which supports the main role of the photochemical process in aptamer immobilization. 

In conditions of light or thermal activation, the azide functionality dissociates into molecular nitrogen and a highly reactive singlet nitrene species, which can chemically bind to graphene [15]. Upon light activation (310 nm), the azide group is converted to the highly reactive singlet form that can subsequently undergo C=C addition reactions with the neighboring graphene sp^2^ C network to form the aziridine adduct [27]. The nitrene is able to attack one of the present carbon double bonds of the graphene by pericyclic [2 + 1] cycloaddition (Figure 1c), which results in the formation of an aziridine ring [28]. Compared to the pyrene-based aptamer immobilization technique, where probe density is defined by the linker density on the surfaces [12], the use of photochemistry may enable the direct modulation of probe density by changing the concentration of aptamers. 

Raman spectroscopy provides information about structure modifications on graphene upon the covalent binding of azide-modified aptamers, such as the decrease in the intensity of the G band (Figure 2a). This change is more pronounced in comparison to the decrease in peak intensity observed upon exposure of pristine graphene to UV light, which occurs due to partial oxidation under UV irradiation (Appendix A). As the covalent attachment of azide to the carbon lattice does not disrupt the crystal structure, the Raman shift cannot be utilized to provide direct proof of covalent reaction [29]. Graphene modification by the azide-aptamer forms a three-membered aziridine-ring linkage with two neighboring carbon atoms, generated by the photochemical activation of organic azides [30]. In this reaction, the sp^2^ hybridization of C-C bonds and electron mobility is preserved (Figure 1c).

AFM was used to characterize the graphene channel before and after attachment of the aptamer. The thickness of the graphene of 4.1 ± 0.8 nm is in line with previously reported results for wet-transferred CVD single-layer graphene because of the presence of trapped water and polymer residuals. The change in thickness that could arise from organic groups and/or water cannot be observed with certainty in the measurements of height changes (Figure 2b). The decrease in graphene surface roughness from 0.67 ± 0.02 nm to 0.56 ± 0.03 nm (Figure 2c,d) may be occurring due to both the partial UV cleaning of residuals and the smoothing of the surface by the aptamer-backbone adsorption.

The changes in the main characteristics of aptamer-modified graphene in comparison to the pristine graphene, irradiated under the same conditions but without aptamers, are provided in Table 1. The small increase in conductivity of the Sensor GFET can be explained by the process of electron doping from the nitrene group, which competes and compensates for the decrease in conductivity that occurs in the process of oxidation of the pristine GFET. This effect of oxidation is clearly visible in the case of decreased conductivity of the Control GFET. The more prominent shift of the Dirac point in the case of the Sensor GFET (48 mV) in comparison to the Control GFET (17 mV) can be mainly attributed to the effect of the aptamer, which brings negative charges close to the graphene surface. The intensity of Raman G band decreases to ca. 1/2 of the original intensity in the case of the Sensor GFET, while in case of the Control GFET the intensity is lowered only to 1/3 of the original intensity, which also evidences the successful modification of the Sensor GFET surface.

Comparison of the EDX data (Table 2, Appendix A) does not lead to unambiguous evidence of surface functionalization, as the sensitivity of EDX to the organic groups is very low [31]. Nevertheless, presence of the nitrogen band at 392 eV can be clearly observed after binding of the azide-modified aptamer, which is absent in the case of pristine graphene spectrum. The summarized EDX data are collected in Table 2.

### 3.2. NT-proBNP Measurements by Photochemically Modified Graphene Sensor

Measurements of current–voltage characteristics at different concentrations of NT-proBNP (Figure 3a) revealed a noticeable Dirac point shift of GFETs under exposures to different NT-proBNP concentrations (Figure 3b). Noticeably, no change in transconductance was observed. Saturation of the Dirac point shifts was observed at a concentration of ~10^3^ pg/mL, while linear dependence was observed in the range from 0.1 pg/mL to 100 pg/mL. As there were no changes in the mobility of the main charge carriers, the similar current change was observed both for the hole (at V_G_ = 100 mV) (Figure 3c) and electron (V_G_ = 350 mV) curve branches (Figure 3d). The total change in current of more than 30% is observed throughout the six orders of magnitude of NT-proBNP concentration. The measurement of NT-proBNP was carried out in 0.1 × PBS and Twin solution to increase the Debye length [19].

We further applied the model for the Langmuir isotherm for binding between protein from the solution and the aptamer receptor on the sensor surface, suggested in [19] to calculate the dissociation constant. Surface coverage (Θ) was plotted as:(1)$$\Theta =\frac{I_{D}\left(C_{0}\right)}{I_{D\;}\left(C_{\propto }\right)}=\frac{C_{0}/K_{D}}{1+C_{0}/K_{D}}$$
where I_D_(C_0_) is a drain current at concentration C_0_, I_D_(C_∞_) is a drain current at an infinite concentration of analyte. The dissociation constant was estimated for both electron and hole regimes and was found to be equal to K_D_ = 30 pg/mL (Figure 3c,d), which proves the high stability of the protein interaction with the GFET sensors.

The demonstrated dynamic range for NT-proBNP is well fitted within the risk stratification range for heart failure of 0.5 ng/mL to 6 ng/mL [32]. Its typical concentration for healthy people under 75 years is 125 pg/mL [33], while the levels of NT-proBNP in blood >450 pg/mL are highly suggestive of heart failure, and the levels <300 pg/mL could rule out acute heart failure [34,35]. Hence, the developed sensor would be applicable for on-site monitoring of cardiac disease markers in blood and for early diagnosis of potential heart disease. Moreover, higher sensitivity of the sensor is beneficial, as it would be capable of operating in other body fluids where the concentration of CM is orders of magnitude lower, such as the NT-proBNP range of 2–10 pg/mL that was determined in the saliva of patients [36]. 

Comparison between the sensor response of aptamer-functionalized vs. nonfunctionalized GFETs is given on Appendix A. In case of the aptamer-functionalized GFET, a sensor response in the linear range of four orders of magnitude of concentrations of NT-proBNP was observed. The increase in pristine graphene response is observed only for concentrations higher than 10^4^ pg/mL, which occurs probably due to adsorption of proteins on graphene surface and direct charge transfer. We also tested the specificity of the sensor, by measuring the response to cTnI, which is also considered as a heart failure marker. Exposure of the sensor to a high concentration of this protein (3.5 × 10^3^ pg/mL) did not produce a significant sensing response (Appendix A). Upon addition of NT-proBNP in a final concentration of 10^3^ pg/mL to the same solution of cTnI, the response of the sensor was again noted. The response of the sensor to the NT-proBNP and cTnI mix is lower compared to the response to pure analyte solution, possibly due to nonspecific surface interactions with biomarkers. 

Finally, we investigated the dynamics of sensor response by comparing the signal intensities upon exposure to the solutions of NT-proBNP in PBS at concentrations of 50 pg/mL and 5 × 10^3^ pg/mL (Figure 4). For the two concentrations of the protein, we observed similar dynamics. However, in conditions of higher concentration of the analyte, the response time of the sensor was shorter, due to the increased probability of binding reactions. 

Hence, the results imply that the photochemically modified, aptamer-functionalized GFET enables high efficiency in NT-proBNP detection, both demonstrating high sensitivity with the limit of detection (LOD) of 0.01 pg/mL (Appendix A) and fast response time, which is promising for POC applications. A comparison of the performances of the biosensors from this work and the available studies from the literature is provided in Table 3.

## 4. Conclusions

The results indicate that after exposure to UV light, the azide-modified aptamers were successfully bound to the GFET surface. The produced functionalized GFET-based sensor was demonstrated for the detection of NT-proBNP in a wide range of concentrations from 10^−1^ to 10^4^ pg/mL, which is the range that overlaps with the risk stratification range for heart failure. The developed aptabiosensor provides the lowest limit of detection and response time compared to the other field-effect transistor-based sensors to NT-proBNP from the literature.

The functionalization methodology is based on a simple photochemical process, which, despite possible partial UV-induced damage to the aptamers, still enables graphene functionalization and the construction of GFET sensors with clinically relevant sensitivity. The one-step photochemical methodology of graphene surface functionalization may show promising attributes in terms of scalability and compatibility with CMOS processes for graphene device production. The methodology could be applicable for the construction of different graphene-based sensors in the future, as well as for the possible maskless light-assisted patterning of multiple receptors on a single chip, paving the way for the development of multiplex sensors.

## Figures and Tables

**Figure 1 biosensors-12-01071-f001:**
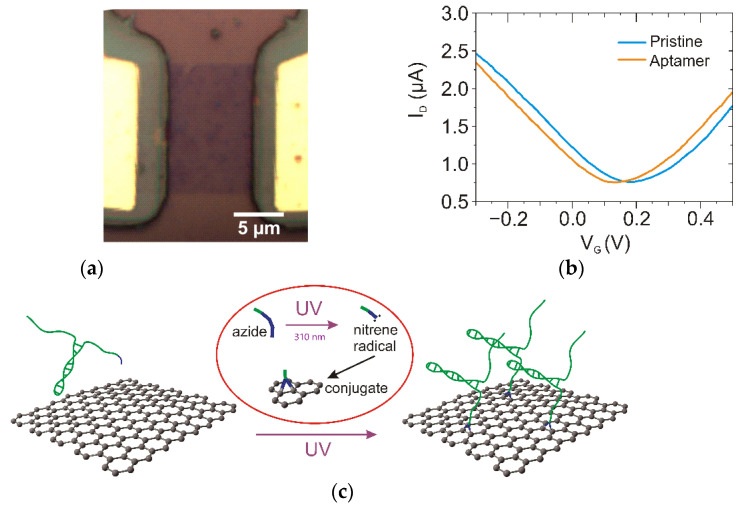
Photochemical immobilization of aptamer on graphene channel of FET. Optical image of GFET channel with passivated gold electrodes (**a**). Change in I-V_G_ curves for GFETs modified with azide-aptamer (V_DS_ = 100 mV) (**b**). (**c**) Scheme of photochemical binding azido-modified aptamer to graphene: azide group under 310 nm UV light converts to nitrene radical, which covalently binds to graphene by pericyclic [2 + 1] cycloaddition.

**Figure 2 biosensors-12-01071-f002:**
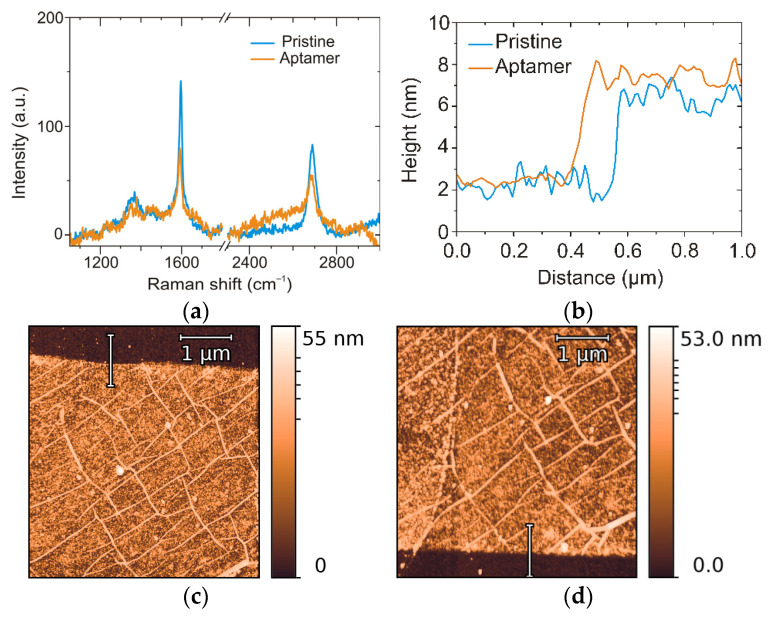
Characterization of the sensor surface after photochemical attachment of the aptamer. Raman shifts of graphene before and after azide-aptamer immobilization (**a**). Cross section of AFM image of graphene channel before and after aptamer binding (**b**). AFM images for pristine graphene channel (**c**), and channel after azide-modified aptamer attachment (**d**). Lines on (**c**,**d**) show the cross-section positions for (**b**).

**Figure 3 biosensors-12-01071-f003:**
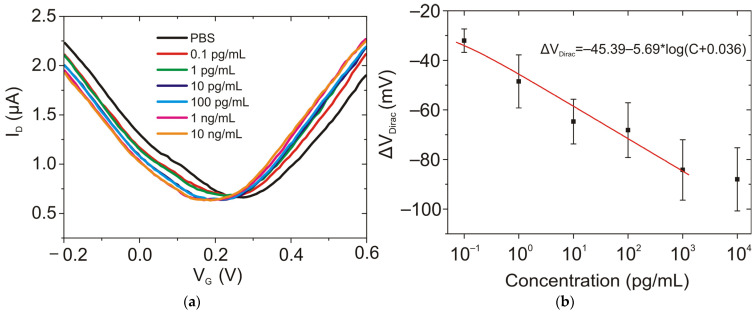
The GFET sensor response to NT-proBNP in 0.1 × PBS (pH 7.4). (**a**) Transfer curves after stabilization for each NT-proBNP concentration (0.1, 1, 10, 100, 1000, and 10,000 pg/mL) in 0.1 × PBS with no washing steps, with applied V_DS_ = 50 mV. (**b**) Change in Dirac point for increasing NT-proBNP concentration. (**c**) I_D_ change for each NT-proBNP concentration at a fixed gate voltage V_G_ = 100 mV with respective Langmuir binding isotherm. Inset: the linear range for low-concentration response. (**d**) I_D_ change for each NT-proBNP concentration at a fixed gate voltage V_G_ = 350 mV with respective Langmuir binding isotherm. Inset: the linear range for low-concentration response.

**Figure 4 biosensors-12-01071-f004:**
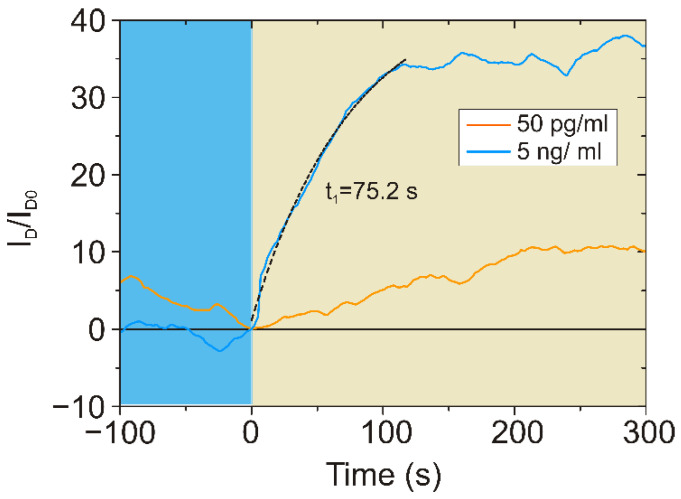
Time-dependent measurement changes in relative drain current I_D_ for different concentrations of NT-proBNP in 0.1 × PBS, without regeneration, V_G_ = 100 mV and V_DS_ = 100 mV. The response time *t*_1_ was calculated in single-exponent approximation.

**Table 1 biosensors-12-01071-t001:** The change in resistance (R), Dirac point, and G band intensities (I) after photochemical modification of graphene channel.

Parameter	Sensor GFETs *	Control GFETs **
R before UV, kOhm	390 ± 130	470 ± 200
R after UV, kOhm	360 ± 200	560 ± 140
Dirac point before UV, V	0.183 ± 0.004	0.261 ± 0.004
Dirac point after UV, V	0.135 ± 0.004	0.244 ± 0.004
Raman I(G) before/I(G) after	1.97	1.48

* Sensor GFET—GFET modified with azide-aptamer under UV, N = 5; ** Control GFET—GFET after UV irradiation, N = 4.

**Table 2 biosensors-12-01071-t002:** EDS data on graphene with and without azide-modified aptamer.

Element	Sensor GFETs	Control GFETs
C	3.6 ± 1.4	3 ± 1
N	0.52 ± 0.45	0.45 ± 0.31
O	57 ± 13	58 ± 11
Si	39 ± 7	39 ± 3

**Table 3 biosensors-12-01071-t003:** FET-based sensors for NT-proBNP.

FET Type	Receptor Type	Response Time	LOD	Detection Range	Ref
GFET	aptamer	75 s	0.01 pg/mL	0.1–10^4^ pg/mL	This work
rGO-FET	antibody	15 min	30 pg/mL	1–10^4^ pg/mL	[18]
AlGaN/GaN HEMT	aptamer	5 min	220 pg/mL	10^2^–10^4^ pg/mL	[8]
AlGaN/GaN HEMT	antibody	-	1 pM (~10 pg/mL)	0.1–10^3^ pM	[37]
Si(n+)/SiO_2_/Si_3_N_4_	antibody	-	0.02 pg/mL	10–2 × 10^4^ pg/mL	[38]

## Data Availability

The data presented in this study are available on request from the corresponding authors.

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
