# Peer review of "One-Step Photochemical Immobilization of Aptamer on Graphene for Label-Free Detection of NT-proBNP"

_biosensors, 2022, doi:10.3390/bios12121071_

Round 1

Reviewer 1 Report

Some points of the manuscript should be improved. Specific comments are given below.

1) Please carefully check the text for writting and grammar. The text contains some mistakes. 

2) The limit of detection (LOD) included in the abstract is different to the limit of detection showed in the Table 3. The FET-based sensors showed in the Table 3 should be compared with the developed aptabiosensor according their analytical parameters.

4) The units should be unified in the manuscript. See line 258.

5) The authors should be comment the advantages of the developed aptabiosensor. 

6) The authors should not write sentences longer than three lines if they want readers to understand and cite their work. 

7) The size of text labels at x-axes and y-axes of plots typically should be drawn in >32 point size. Some labels in graphs are too small and should be made larger in order to improve legibility. 

8) All graphs in the manuscript and electronic suppelmentary information should be explained in great detail (by captions and explanatory text). 

Reviewer 2 Report

Overall, this manuscript provides an approach based on one-step photochemical immobilization of aptamer on graphene for label-free detection of a heart failure biomarker (NT-proBNP). This paper provides a simple and practical UV-based method for DNA immobilization and biosensor functionalization, which can be developed as a universal method. In addition, the developed GFTE biosensor can detect NT-proBNP with a low LOD and short response time. This is interesting work that is worth of publication in Biosensors. However, the following concerns should be addressed before publication.

Major concerns:

1. The immobilization of azide-modified DNA aptamer using 310 nm UV is a good approach. However, there are other UV-free immobilization strategies such as biotin-streptavidin. Compared to those traditional modifications, what is the advantage of this photochemical approach? In addition to azide, diazirine-based photocrosslinking which is triggered by 365 nm UV can also meet the requirement of immobilization. The authors should fully discuss that.

2. In Figure 1b, a DNA aptamer without azide should be tested as negative control, showing that the immobilization was not due to the non-specific physical adsorption.

3. A key factor that affects the performance of the biosensor is the probe coverage on the chip surface. What is the estimated grafting density of the DNA aptamer? Was there any optimization on the grafting density?

4. The author should discuss the linear range of GFTE while detecting NT-proBNP. A standard curve needs to be established, especially in a range of 0.5~6 ng/mL, so that the readout can be quantified for POC.

Minor concerns:

1. Please revise the abstract. GFTE is the key concept of the whole paper, so it’s better to highlight it in the abstract.

2. Please check if there any missing/wrong abbreviation (e.g., Page 2, Line 89: Atomic force microscopy). Please add full name of the abbreviation if it is the first time to introduce the concept (e.g., NT-proBNP in abstract).

3. Please revise all the format of the numbers and units in the manuscript, and keep them identical (e.g., ml vs mL; 0,45 vs 0.45, pbs vs PBS, etc).

4. In Figure 1c, the author should add “310 nm” to highlight the wavelength of UV used in this research, since this is one of the key points for the immobilization strategy.

Round 2

Reviewer 2 Report

The revised the manuscript has already addressed the concerns in the previous review comments. The description on the data matches the experimental design. In addition, more details have been added in the results section, which provides more comprehensive discussion. Overall, the revised manuscript is suitable for publication in Biosensors. Please carefully check the spelling and figures before submitting the final version.